# Genetic Dissection of Diverse Seed Coat Patterns in Cowpea through a Comprehensive GWAS Approach

**DOI:** 10.3390/plants13091275

**Published:** 2024-05-05

**Authors:** Haizheng Xiong, Yilin Chen, Waltram Ravelombola, Beiquan Mou, Xiaolun Sun, Qingyang Zhang, Yiting Xiao, Yang Tian, Qun Luo, Ibtisam Alatawi, Kenani Edward Chiwina, Hanan Mohammedsaeed Alkabkabi, Ainong Shi

**Affiliations:** 1Department of Horticulture, University of Arkansas, Fayetteville, AR 72701, USA; yc046@uark.edu (Y.C.);; 2Texas A&M AgriLife Research, 11708 Highway 70 South, Vernon, TX 76384, USA; 3Sam Farr U.S. Crop Improvement and Protection Research Center, U.S. Department of Agriculture, Agricultural Research Service (USDA-ARS), Salinas, CA 93905, USA; 4Department of Poultry Science & The Center of Excellence for Poultry Science, University of Arkansas, Fayetteville, AR 72701, USA; 5Mathematical Sciences, University of Arkansas, Fayetteville, AR 72701, USA; 6Biological Engineering, University of Arkansas, Fayetteville, AR 72701, USA; 7Program of Material Science and Engineering, Fayetteville, AR 72701, USA

**Keywords:** cowpea, GWAS, seed coat pattern, seed coat color

## Abstract

This study investigates the genetic determinants of seed coat color and pattern variations in cowpea (*Vigna unguiculata*), employing a genome-wide association approach. Analyzing a mapping panel of 296 cowpea varieties with 110,000 single nucleotide polymorphisms (SNPs), we focused on eight unique coat patterns: (1) Red and (2) Cream seed; (3) White and (4) Brown/Tan seed coat; (5) Pink, (6) Black, (7) Browneye and (8) Red/Brown Holstein. Across six GWAS models (GLM, SRM, MLM, MLMM, FarmCPU from GAPIT3, and TASSEL5), 13 significant SNP markers were identified and led to the discovery of 23 candidate genes. Among these, four specific genes may play a direct role in determining seed coat pigment. These findings lay a foundational basis for future breeding programs aimed at creating cowpea varieties aligned with consumer preferences and market requirements.

## 1. Introduction

Cowpea (*Vigna unguiculata* [L.] Walp.), a diploid warm-season legume (2*n* = 22), plays a pivotal role as a primary source of protein and calories, particularly in sub-Saharan Africa, the Mediterranean Basin, Southeast Asia, Latin America, and the United States [1,2]. In 2022, over 9.8 million metric tons of dry cowpeas were reported globally, which usually grow as an intercrop with maize, sorghum, or millet. Due to its adaptability to heat and drought, along with its ability to fix nitrogen, cowpea is a versatile and resilient crop [3]. Beyond its nutritional significance, the cowpea’s aesthetic allure lies in the diversity of seed coat colors, ranging from deep blacks to vibrant browns [4,5]. Seed coat pattern/color is an important aspect of breeding strategies. In both agricultural and consumer contexts, cowpea seed coat holds significance [6]. Breeders prioritize patterns such as Pinkeye, Blackeye, and Browneye in commercial breeding programs because specific colors appeal to different markets; consumer preferences, heavily influenced by the seed coat, significantly affect their purchasing and consumption decisions [7,8]. Different regions have distinct preferences; for example, West Africa values traits like a lack of color for flour or solid brown for whole beans [4], while in the United States, “black-eyed peas” with tight black eyes are preferred [9]. Investigations into the genetic basis of cowpea seed coat color and pattern offer crucial insights for enhancing techniques, practices, and processes of cowpea breeding [10,11]. Such research equips breeders with the knowledge to cultivate varieties that meet specific preferences, while also delving into the genetic underpinnings of color variation in cowpeas [12].

The range of colors observed in cowpea seeds results from various developmental processes, which reveal the underlying factors responsible for the striking palette of hues [13]. Earlier investigations by Spillman and Harland in the 20th century successfully identified the genetic factors accountable for color expression in cowpea seeds [14]. These primary factors, referred to as “Color Factor” (C), “Watson” (W), and “Holstein” (H), significantly contribute to the intriguing diversity found in seed coat patterns [6]. To enhance our understanding of how different color traits are inherited in cowpea seeds, researchers have extensively studied cross ability and the inheritance of seed coat color. These investigations have unraveled the mechanisms through which various color traits segregate in the progeny [15,16,17]. Mustapha’s comprehensive study (2009) elucidates the inheritance of seed coat color pattern by hybridization experiments within three pairs of parental crosses, and it also revealed the involvement of multiple, possibly allelic genes in controlling these color patterns [4]. In Ajayi’s study (2020) using two cowpea accessions, IT98K-205-8 (white seeded) and IT98K-555-1 (brown seeded), it was found the polygenic inheritance and epistasis in seed coat color exhibit maternal effects influencing cross ability [15]. Gaafar’s research revealed that gamma radiation effectively induced genetic variations in an Egyptian cowpea cultivar, leading to black seeds in the M2 generation and further changes in M3, including variations in seed coat color and eye pattern [16]. In 2014, Herniter conducted a study estimating the genes responsible for seed coat color in cowpea; in 2019, he introduced a model with six stages of seed coat development, aiding the comprehension of observed phenotypes [18]. These resources encompass a comprehensive reference genome sequence and genotyping arrays, incorporating numerous single nucleotide polymorphisms (SNPs) [19]. Leveraging these resources, researchers are trying to gain substantial insights into the precise genetic mapping of seed coat traits in cowpea seeds.

Our study employs a GWAS approach utilizing a 296-cowpea mapping panel and 110 K SNPs to identify markers and candidate genes associated with a spectrum of coat colors and patterns, including two full seed coats: Red and Cream, two prime seed coats: White, Brown/Tan, and four seed patterns: Blackeye, Pinkeye, Browneye, and Red/Brown Holstein (Figure 1). This research contributes to the broader body of knowledge on cowpea genetic diversity, offering new avenues for the enhancement of cowpea varieties in response to both agronomic and market demands.

## 2. Results

### 2.1. Identification of Seed Coat Patterns

The visual examination of seed coat color/pattern was conducted every year from 2014 to 2020 in two locations. Phenotypic data for each accession in the observed populations can be found in Appendix A. The classification of accessions revealed that 21 were characterized by the Red (full coat), 102 as the White coat, 82 as the Tan/Brown coat, 36 as the Blackeye, 27 as the Pinkeye, 45 as the Browneye, 20 as the Cream (full coat), and 24 as the Red/Brown Holstein.

### 2.2. The Genotyping, Diversity and Structure Analysis

In the present study, a comprehensive set of 110 K high-quality single-nucleotide polymorphisms (SNPs) was employed for the Genome-Wide Association Study (GWAS). The average minor allele frequency (MAF) across the entire genome was observed to be 22.1%. Additionally, the rates of heterozygosity and missingness were determined to be 2.2% and 0.3%, respectively. Notably, the average inter-SNP distance exhibited variation across the different chromosomes, ranging from 3.4 kilobases (kb) on Chromosome 2 to 6.5 kb on Chromosome 3, with an overall mean of 4.3 kb.

The population structure and admixture dynamics of 296 cowpea accessions were elucidated employing the “LEA” package within the R statistical environment [20]. This analysis, grounded on 110 K high-fidelity Single Nucleotide Polymorphisms (SNPs), revealed a notable peak in delta K at K = 3, indicative of three distinct subpopulations within the dataset. Utilizing a designated threshold of 0.45 for population assignment, the distribution among the subpopulations was as follows: 64 accessions (21.6%) were allocated to the Q1 subpopulation, 91 accessions (30.7%) to Q2, and 134 accessions (45.3%) to Q3. A residual six accessions were categorized within a mixed group, as detailed in Figure 2A and Appendix A. Concordance with these findings was observed in the Principal Component Analysis (PCA), which also delineated the three subpopulations (Figure 2B). Further, the Neighbor-Joining phylogenetic analysis corroborated these groupings, with the most genetically similar accessions clustering within adjacent branches of the phylogenetic tree (Figure 2C). The kinship, which shows the genetic relationships among accessions, is shown in Appendix A. This provides further information into the population’s genetic structure as required by the GWAS model.

### 2.3. LD in Cowpea Genome

We examined the patterns of genome-wide linkage disequilibrium (LD) between the SNP markers mapping panel of our cowpea collection for identifying the genetic basis of GWAS in cowpea. Overall, we found that LD decays to R^2^ = 0.80 by 35.6 kb and R^2^ = 0.20 by 79.7 kb (Appendix A) on average. 

### 2.4. GWAS Analysis and Candidate Genes

The GWAS examined 110 K SNPs using four models (BLINK, FarmCPU, MLMM, and MLM) in GAPIT 3, and three models (SMR, GLM, and MLM) in TASSEL 5. We maintained a consistent significance threshold (−log(P) ≥ 6.34) across all models and prioritized SNP consistency among them. SNPs significant in over four of the seven models were deemed significant. We identified 18 SNPs associated with eight seed coat patterns, each showing the significance in at least four models. The qualified SNPs for each model (GAPIT3) are in Appendix A, and Manhattan plots (TASSEL 5) are in Appendix A, underlining result coherence across all models.

#### 2.4.1. Blackeye 

We identified two SNPs associated with the Blackeye pattern, located on chromosomes Vu01 and Vu05 at positions 26,234,865 and 3,015,637, respectively (Figure 3). Particularly noteworthy is the SNP marker Vu05_3015637, which demonstrated the highest logarithm of odds (LOD) value of 46.2 according to the BLINK model and exhibited a LOD > 9.0 across five models. Additionally, it was found to explain up to 27.2% of the phenotypic variance (PVE). The SNP Vu01_26234865 displayed a moderately high LOD of 16.6 within the BLINK model, with a PVE of 10.5%.

Within a genomic region spanning 10 kb from the two identified SNPs linked to the Blackeye pattern on chromosomes Vu01 and Vu05 (as detailed in Table 1), three candidate genes emerged. These genes were annotated as follows: the MATE efflux family protein, recognized for its pivotal role in facilitating the efflux of secondary metabolites in plants; the Cyclic nucleotide-regulated ion channel family protein, known for its involvement in various physiological processes; and the Calcium/lipid-binding (CaLB) phosphatase, which contributes to diverse physiological functions.

#### 2.4.2. Browneye 

Two SNPs linked to the Browneye pattern were identified on chromosomes Vu03 and Vu10, specifically at positions 17,993,754 and 38,465,180, respectively (Figure 4). Of particular significance is the SNP marker Vu10_38465180, which exhibited the highest LOD of 14.8 according to the BLINK model and maintained an LOD > 8.7 across four models. Moreover, this SNP was found to account for up to 15.3% of the PVE. The SNP Vu03_17993754 displayed a moderately high LOD of 7.9 within the FarmCPU model, with a PVE of 6.0%.

Within a span of 10 kilobases from the two identified SNPs associated with the Browneye pattern on chromosomes Vu03 and Vu10 (as delineated in Table 2), two candidate genes emerged: *Vigun03g161700* and *Vigun10g165500*. Notably, *Vigun03g161700* was characterized as a member of the PPR superfamily protein, known for its role as RNA-binding proteins crucial in RNA metabolism. Meanwhile, *Vigun10g165500* was identified as the Transcription Initiation Factor TFIID Subunit A, essential for regulating the transcription of a diverse array of genes involved in various biological processes.

#### 2.4.3. Pinkeye

Two SNPs associated with the Pinkeye pattern were identified on chromosomes Vu05 and Vu10, specifically at positions 3,015,637 and 38,465,180, respectively (Figure 5). Notably, the SNP marker Vu10_38465180 exhibited the highest LOD of 15.6 according to the FarmCPU model and consistently retained an LOD > 7.0 across four models. Additionally, this SNP was found to explain up to 32.6% of the PVE. The SNP Vu05_3015637 displayed a moderately high LOD of 8.4 within the BLINK model, with a PVE of 21.6%.

Within 10 kb from the two significant SNPs linked to the Pinkeye pattern (as outlined in Table 3), two candidate genes (*Vigun05g037200* and *Vigun05g037300*) were discerned by SNP Vu05_3015637 and annotated as follows: Cyclic nucleotide-regulated ion channel family protein and Calcium/lipid-binding (CaLB) phosphatase, respectively. Additionally, on chromosome Vu10, the gene *Vigun10g165500* (transcription initiation factor TFIID subunit A) was identified by SNP Vu10_38465180.

#### 2.4.4. Red/Brown Holstein

Three SNPs linked to the Holstein pattern were identified on chromosomes Vu08, Vu09, and Vu10, specifically at positions 4,097,874, 29,253,230, and 39,484,624, respectively (Figure 6). Remarkably, the SNP marker Vu08_34097874 displayed the highest LOD of 23.2 according to the BLINK model, consistently maintaining a LOD > 6.6 across five models. Furthermore, this SNP elucidated up to 14.0% of the PVE. The SNPs Vu09_29253230 and Vu10_39484624 exhibited moderately high LOD values of 18.1 and 15.0, respectively, along with their respective PVE, as determined by the FarmCPU model.

Within a 10 kb proximity of the identified SNPs linked to the Holstein pattern (as detailed in Table 4), specific genomic regions were pinpointed. The SNP Vu08_34097874 resides in intergenic regions between *Vigun08g170100* and *Vigun08g170200*, both annotated as members of the GDSL-like Lipase/Acylhydrolase superfamily. These proteins are known for their involvement in lipid metabolism, plant defense, and stress responses. The SNP Vu09_29253230 is situated downstream of *Vigun09g133400*, characterized as a Nucleic acid-binding, OB-fold-like protein, which plays diverse roles in nucleic acid metabolism. Finally, the Vu10_39484624 SNP is positioned amidst three candidate genes: *Vigun10g176200*, *Vigun10g176300*, and *Vigun10g176400*. These genes encode proteins belonging to the Amino acid dehydrogenase family, the Ubiquitin-conjugating enzyme family, and Tudor/PWWP/MBT superfamily, respectively. They are involved in nitrogen metabolism and stress responses, protein degradation and cellular regulation, and regulation of chromatin structure, gene expression, and plant development, respectively.

#### 2.4.5. Cream

Two SNPs associated with the Cream pattern were identified on chromosomes Vu07 and Vu11, specifically at positions 21,041,365 and 1,810,109, respectively (Figure 7). Notably, the SNP marker Vu07_21041365 displayed a high LOD of 79.3, consistently maintaining a LOD > 15.6 across six models. Additionally, it exhibited a PVE of 17.0% in the BLINK model. The SNP Vu11_1810109 demonstrated a high LOD of 36.9 according to the BLINK model and retained an LOD > 7.1 across six models.

Within a 10 kb vicinity of the identified SNPs linked to the Cream seed coat on chromosomes Vu07 and Vu11 (as detailed in Table 5), two candidate genes emerged: *Vigun07g113700* and *Vigun11g014700*. These genes were annotated as a glycosyl transferase 8 domain-containing protein and a glycoside hydrolase, respectively. The glycosyl transferase 8 domain-containing protein is involved in synthesizing diverse glycoconjugates crucial for plant growth, development, and stress responses. Meanwhile, the glycoside hydrolase plays pivotal roles in carbohydrate metabolism, cell wall modification, and defense against pathogens in plants.

#### 2.4.6. Brown/Tan Coat

Two SNPs associated with the Brown/Tan coat were identified on chromosomes Vu05 and Vu08, specifically at positions 3,137,965 and 36,618,860, respectively (Figure 8). Remarkably, the SNP marker Vu08_36618860 displayed the highest LOD of 21.3 according to the SMR model, while Vu05_3137965 exhibited a moderately high LOD of 11.0 in the FarmCPU model.

Within a 10 kb vicinity of the identified SNPs associated with the Brown/Tan pattern on chromosomes Vu05 and Vu08 (as detailed in Table 6), three candidate genes were pinpointed: *Vigun05g039300*, *Vigun08g201900*, and *Vigun08g202000*. These genes were annotated as Myb domain protein 114, PYRIMIDINE B, and co-factor for nitrate, reductase, and xanthine dehydrogenase 5, respectively. Among these genes, Myb domain protein 114 has been notably associated with the regulation of various aspects of plant development, particularly in seed coat formation and pigmentation.

#### 2.4.7. White Coat

Two single SNPs linked to the White coat were identified on chromosomes Vu07 and Vu10, positioned at 21,041,365 and 38,465,180, respectively (Figure 9). Of notable significance is the SNP marker Vu10_38465180, which displayed the highest LOD of 50.3 according to the BLINK model, consistently maintaining a LOD > 16.3 across all models. Moreover, it elucidated up to 25.5% of the PV. The SNP Vu07_21041365 exhibited a moderately high LOD of 9.4 within the BLINK and FarmCPU models.

Within 5–10 kb proximity from the two identified SNPs associated with the White seed coat on chromosomes Vu07 and Vu10 (as outlined in Table 7), two candidate genes were discerned. These genes were annotated as glycosyl transferase 8 domain-containing protein and Transcription initiation factor TFIID subunit A.

#### 2.4.8. Red Seed

Three SNPs linked to the Red seed phenotype were detected on chromosomes Vu03, Vu04, and Vu09, specifically at positions 5,249,749, 2,848,654, and 38,749,459, respectively (Figure 10). Notably, the SNP marker Vu08_36618860 exhibited the highest LOD of 50.7 according to the MLMM model, consistently maintaining an LOD > 6.7 across six models. Additionally, Vu04_2848654 and Vu09_38749459 displayed high LOD values of 32.2 and 28.6, respectively, in the BLINK model.

Within a 10 kb range of the identified SNPs associated with the Red seed coat on chromosomes Vu03, Vu04, and Vu08 (as outlined in Table 8), seven candidate genes were identified: *Vigun03g063900*, *Vigun03g064000*, *Vigun03g064100*, *Vigun04g034400*, *Vigun04g034500*, *Vigun09g213300*, and *Vigun09g213400*. These genes play pivotal roles in various biological processes, including lipid metabolism, metabolic pathways, plant development, cell wall synthesis, and cellular signaling.

## 3. Discussion

The research on cowpea seed coat patterns is both complicated and interesting due to its intricate genetic basis and significant implications for agriculture and food science [21]. The complexity arises from the seed coat development process, where the integuments of the ovule differentiate into specialized cell types, involving a sophisticated network of genetic and environmental interactions [22,23]. This process is controlled by multiple genes, as demonstrated in studies identifying QTL associated with seed coat patterns and colors in cowpea. Further complicating the research is the discovery that seed coat pigmentation patterning in cowpea is governed by a multi-locus system, suggesting a complex genetic architecture underlying these traits [4,6,15,18]. 

The three-locus system controlling seed coat patterns in cowpea is a classical framework that has been established through extensive research. This system involves the interaction of three loci, traditionally identified as the Color Factor (C), Watson (W), and Holstein (H) factors [6,16,24]. These loci work together to determine the diverse patterns and colors observed in cowpea seed coats, such as the distribution and intensity of pigmentation. On this basis, we also refer to the USDA classification method of “main color + pattern type + pattern color” to define the main seed-coat types in the population [25]. Certain studies have utilized the dimensions of seed coat patterns or the uniformity of seed coat color as classification criteria, presenting a methodological approach that helps further exploration in academic contexts [26,27]. 

### 3.1. Genetic Diversity of the Mapping Population

The detailed morphological evaluation of cowpea traits conducted from 2014 to 2020 across two distinct locations provides significant insights into the genetic diversity of cowpea accessions. This analysis, which includes a broad spectrum of phenotypic traits such as plant height and type, leaf shape, pod position and placement, flower color, seed weight, and texture, reflects the extensive genetic variations present within the mapping population in our former studies [28,29,30]. Although the specific observations are not presented in this study, the collected phenotypic data underscore the variability and serve as a foundation for understanding the underlying genetic mechanisms governing these traits. The investigation into cowpea morphology not only enhances our comprehension of the species’ genetic complexity but also facilitates the identification of specific traits that could be targeted for improvement through breeding programs, thereby contributing to the advancement of cowpea as a crop [31].

The employment of 110 K high-fidelity SNPs offers a high-resolution genetic landscape of cowpea, facilitating the identification of genetic variants associated with specific phenotypic traits [22,32,33,34,35]. The revelation of three distinct subpopulations enables researchers to control population stratification. The concordance of these findings with the results from PCA and Neighbor-Joining phylogenetic analysis not only validates the subpopulation structure but also provides a visual and genetic framework for understanding the relationships among accessions [36]. This comprehensive approach to delineating genetic relatedness and structure within the population is invaluable for GWAS, as it aids in the identification of genetic markers linked to desirable traits [37]. Moreover, the kinship matrix, which delineates the genetic relatedness among accessions, offers additional insights that are crucial for GWAS. Understanding the degree of relatedness between individuals helps to further adjust the analysis for background genetic correlations, ensuring that associations between SNPs and traits are not confounded by genetic relatedness [38].

### 3.2. The LD Decay

The research into LD decay in genomes, as summarized by our analysis with an R^2^ = 0.2 and a decay distance of 87.6 Kb, offers a pivotal insight into genetic architecture and association mapping strategies. Our estimation that an associated locus will, on average, encompass 9 genes and 22 SNPs within the reference genome is significant for understanding the potential complexity and genetic diversity within any given region of interest [22,39]. However, it is crucial to note the inherent limitations of these estimates due to the non-uniform distribution of genes and SNPs across the genome and variations in local recombination rates.

This observation aligns with the broader findings across multiple studies, where the LD decay distance can vary significantly across different organisms and even within different regions of the same genome [40,41]. For instance, in diverse populations or species, the average LD decay could be shorter due to higher historical recombination rates [42], or it could extend further in populations with low diversity or those that have undergone a bottleneck [43]. The complexity of genetic architecture highlighted by our analysis emphasizes the importance of applying established methods such as GWAS and genetic mapping in a comprehensive manner. While our estimates provide a useful starting point for identifying associated loci, the actual gene and SNP count in any associated region could be higher or lower, influenced by local genomic features and evolutionary history. This variability emphasizes the importance of incorporating comprehensive genomic information and considering the specific context of the population or species under study when interpreting LD and its implications for genetic research [44].

### 3.3. GWAS Analysis

In the current landscape of genomic research on cowpea, Recombinant Inbred Lines (RILs) and Multi-Parent Advanced Generation Inter-Cross (MAGIC) populations have been predominantly utilized for mapping studies to coat pattern and color. These populations offer specific advantages, including high levels of homozygosity in RILs and enhanced genetic diversity and recombination in MAGIC populations, which facilitate the fine-mapping of quantitative trait loci (QTLs) [6,15,16,17]. In this study, the use of germplasm populations for GWAS in cowpea, particularly for investigating coat pattern and color, represents a novel approach compared to the traditionally utilized RIL or MAGIC. Unlike RILs and MAGIC populations, which are derived from a limited number of parental lines, germplasm populations encompass a broader genetic diversity that represents a wide range of alleles found in natural and breeding populations [45]. This diversity can offer several benefits for GWAS, including the increased allelic variation and population adaptation [46]. The extensive genetic variation in germplasm populations increases the potential for discovering novel genetic associations with coat color and pattern [26]. The loci (Appendix A) identified in our study are consistent with those discovered through QTL mapping in both F2 and advanced RIL populations, demonstrating GWAS’s effectiveness in validating and enriching our knowledge of the genetic underpinnings of seed coat pigmentation [18,27].

Despite these advantages, using germplasm populations for GWAS also comes with challenges, such as controlling for population structure and relatedness, which can confound association signals [47]. Advanced statistical models are required to account for these factors effectively, especially considering the seed coat color and pattern in cowpea are complex and likely influenced by multiple genes. Multiple models can accommodate different types of genetic effects, including additive, dominant, and epistatic interactions, providing a more comprehensive understanding of the genetic architecture underlying these traits [48]. In this study, we applied multiple models: four in GAPIT and three in TASSEL to identify the significant associated SNPs in cowpea coat [49]. The markers which have shown the significance of at least four models were picked to locate the alleles and candidate genes. Those consistent SNPs across different models provide a stronger validation of the genetics [50]. It suggests a robust relationship not dependent on the model’s assumptions or limitations, reducing the risk of false positives, and accounts for the complex genetic architecture influencing these traits [51].

Three genetic markers have been identified as correlating with various cowpea seed coat traits, underscoring a sophisticated level of genetic specificity. These markers include Vu05_3015637, associated with both Blackeye and Pinkeye traits; Vu07_21041365, linked to Cream and White coat colors; and Vu10_38465180, connected to Browneye, Pinkeye, and White coat variations. This discovery aligns with anticipated patterns, revealing that trait-sharing common markers exhibit overlapping phenotypic characteristics, suggesting a potential genetic basis for these associations. For instance, marker Vu05_3015637 might influence traits related to the presence of an “eye” in the seed coat, hinting at a specific gene’s role. Furthermore, the manifestation of Cream seed coats could be interpreted as a polygenic effect involving genes responsible for “White coat” colors combined with genes influencing the absence or presence of an “eye”. Similarly, the predominance of white as the primary seed coat color in Browneye and Pinkeye varieties may indicate a shared genetic foundation. Nonetheless, the possibility of other genetic phenomena such as Pleiotropy and Conserved Genetic Pathways contributing to these observations cannot be discounted without further detailed investigation [52,53,54].

Among 23 candidate genes related to seed coat characteristics, the gene *Vigun01g096800* is linked to Blackeye seeds and encodes for MATE (Multidrug And Toxic Compound Extrusion) efflux family proteins. These proteins function as antiporters, utilizing the electrochemical gradient of protons (H+) across the cell membrane to efflux organic compounds, including secondary metabolites, from the cells [55]. Some studies suggest that MATE transporters are involved in the transport of precursors necessary for the biosynthesis of pigments in the seed coat [56]. For instance, in model plants like Arabidopsis, certain MATE transporters have been identified that specifically modulate the deposition of flavonoids in the seed coat, directly impacting the coloration and protective qualities of the seeds [57]. The gene *Vigun08g170100* is associated with Red/Brown Holstein and codes for GDSL-like Lipase/Acylhydrolase superfamily proteins. These enzymes, characterized by a specific GDSL motif at the active site, act on a variety of molecules including fats, oils, phospholipids, and other complex lipids [58]. In plants, GDSL lipases are in several processes that can directly or indirectly impact the characteristics of the seed coat such as lipid metabolism, morphogenesis and development. Research has demonstrated that GDSL lipases are involved in cuticle formation in seeds and other plant organs. For instance, in Arabidopsis, certain GDSL lipases have been identified that specifically contribute to the formation and maintenance of the cuticle layer, which is integral to the seed coat [59]. *Vigun07g113700*, related to Cream and White coat seeds, encodes a Glycosyl Transferase 8 domain-containing protein. This enzyme belongs to the glycosyltransferases family, using activated sugar donors like UDP-glucose to transfer sugar units to acceptors, thus modifying the properties of these molecules [60]. Some glycosyltransferases are involved in the modification and stabilization of phenolic compounds, which can contribute to pigmentation and UV protection. This is particularly relevant for seed coats as pigmentation can help in protecting the embryo from UV radiation and predation [61]. The gene *Vigun05g039300* codes for Myb domain protein 114 as the candidate gene of the Tan/Brown coat. Myb domain proteins are known to play a significant role in the regulation of anthocyanin biosynthesis, which contributes to the pigmentation in plants, including seed coats [18]. These transcription factors are part of a larger family of genes involved in the control of various physiological and biochemical pathways in plants, including cell cycle regulation, defense, and metabolism [62,63]. In the context of seed coat color, Myb proteins often interact with other transcription factors to activate the expression of genes involved in the synthesis of pigments, leading to the diverse coloration observed in plant seed coats [64].

## 4. Materials and Methods

### 4.1. Plant Material and Phenotype

A 296-cowpea association panel including 35 Arkansas, seven California breeding lines and 254 USDA germplasm collections was categorized by two prime seed coats: White, Brown/Tan; two full coats: Red and Cream; and four patterns: Blackeye, Browneye, Pinkeye, and Red/Brown Holstein. According to USDA descriptions, the classification based on two types of colors is entirely independent. “Prime seed coats” is determined by the dominant color observed in the seeds, whereas “seed pattern” is defined by the pigmentation traits associated with the described pattern. Similar traits have been consolidated and depicted in Figure 1. The cowpea plants were systematically cultivated on an annual basis for the purpose of phenotypic collection spanning the years 2014 to 2020. This cultivation took place along a singular 14-foot-long row, characterized by a row spacing of 3 feet and an approximate plant spacing of 4 inches at Fayetteville (36°4′ N, 94°9′ S), and Alma (35°29′ N, 94°13′ S). Only those accessions in which the seed coat color and pattern have consistently and stably inherited characteristics over these years were used for this study.

### 4.2. DNA Extraction and Whole Genome Sequencing

Genomic DNA was extracted from young cowpea leaves harvested from a single plant. The leaves, once freeze-dried, were pulverized using a Mixer Mill MM 400^®^ (Haan, Germany). Subsequently, DNA extraction was carried out using the CTAB protocol, followed by the addition of DNA (1 mL) buffer to the ground leaves (0.1 g) [65]. This mixture was then centrifuged at 13,000 rpm for 10 min. Protein denaturation was achieved by incorporating 1 mL of chloroform–isoamyl alcohol (24:1) into each sample, after which DNA was precipitated with the addition of 1 mL of isopropanol. To enhance DNA precipitation, the samples were stored at −20 °C overnight. The DNA pellets were subsequently washed with 70% and 90% ethanol solutions and then allowed to air dry. RNA contamination was eliminated by adding 3 µL of RNase to each sample. The DNA was then dissolved in 200 µL of 0.1× TE buffer. The quantity and quality of DNA were assessed using a NanoDrop 200c spectrophotometer (Thermo SCIENTIFIC, Wilmington, DE, USA) and verified on a 1% agarose gel stained with ethidium bromide, respectively.

DNA sequencing was carried out by Novogene (Novogene website, Beijing China). The DNA was fragmented into 350 bp pieces using the Covaris S2^®^ instrument (Covaris, Inc., Woburn, MA, USA). The resulting DNA fragments were then prepared for sequencing using the NEBNext DNA Library Prep Reagent Set for Illumina (BioLabs, Inc., Ipswich, MA, USA), which involved end-repairing the fragments and adding poly-A tails to each. In situ PCR amplification was performed as described in the literature. The sequencing itself utilized the Illumina HiSeq X Ten Series platform (Illumina website, San Diego, CA, USA), achieving an average coverage of 10×. Reads were aligned by the cowpea reference genome [22] using SOAPaligner/soap2, and the SNPs were identified using SOAPsnp version 1.05 [66]. SNPs that were triallelic or had over 5% missing data, as well as SNPs exhibiting more than 10% heterozygosity, were excluded from analysis. Furthermore, a minor allele frequency (MAF) threshold of 5% was applied. For the GWAS analysis, we meticulously selected 10,000 markers from each chromosome that demonstrated the lowest missing data rates, ensuring a robust dataset for our investigation.

### 4.3. Genetic Diversity, Population Structure and Linkage Disequilibrium

The LEA package in the R programming environment offers a powerful platform for analyzing population structure and detecting genomic signatures [20]. It leverages an efficient adaptation of the structure algorithm, adept at identifying distinct clusters (ranging from K = 2 to K = 20) within SNP data. This is facilitated by configuring the analysis to run with 100,000 iterations and a 10,000 iteration burn-in period across 20 replicates for each K value. To ascertain the optimal number of clusters (K), the cross-entropy criterion was employed, providing a statistically rigorous method for selecting K [20]. Following this, accessions were allocated to specific clusters based on a minimum probability threshold of 0.5. The resultant distribution of the 296 accessions across the identified clusters was subsequently illustrated in a bar plot, reflecting their respective cluster assignments. Additionally, phylogenetic relationships and principal component analyses (PCAs) were conducted using TASSEL 5.0 [67] and JMP Genomics software from SAS Institute, Cary, NC, USA. We integrated data from context structure analysis using MEGA 11 software [68] to improve the visualization of phylogenetic trees and PCA plots. Pairwise linkage disequilibrium between markers was also calculated by TASSEL 5.0 with a 200 LD window size.

### 4.4. GWAS and Candidate Genes

Association analyses were conducted using TASSEL 5.0 software [69], applying:

(1) General Linear Model (GLM):

Y = Xβ + ϵ

Y is the vector of observed traits

X is the design matrix for marker genotypes

β is the vector of marker effects

ϵ is the vector of residual errors;

(2) Single-marker regression (SMR): A straightforward approach where each genetic marker is tested individually for its effect on the trait, similar to GLM but typically emphasizing the univariate nature of the test;

(3) Mixed Linear Model (MLM) approaches [70]:

Y = Xβ + Zu + ϵ

Z is the design matrix for random effects

u is the vector of random effects (kinship)

Additional models, including:(1)Multiple Loci Mixed Linear Model (MLMM): MLM integrates multiple loci in the random effects component.(2)Fixed and Random Model Circulating Probability Unification (FarmCPU) [71]: reduces confounding from kinship without using conventional kinship estimates. It selects associated markers through maximum likelihood, based on kinship from these markers alone, preventing overfitting seen in stepwise regression. By alternating between fixed and random effect models, it precisely identifies associated markers, avoiding background genetic noise.(3)Bayesian-information and Linkage-disequilibrium Iteratively Nested Keyway (BLINK) [72]: The random model was replaced by a fixed model in BLINK.
were executed within the R software GAPIT 3 [73], setting the PCA, structural analysis, and kinship estimation to meet the model’s requirements. Bonferroni correction is used to adjust significance. In the study, we determined the desired significance level = 0.05. Adjusted significance threshold (P) = alpha (0.05)/number of SNPs (110 K) ≈ 4.55 × 10^−7^, logarithm of odds = −log(P) ≥ 6.34.

Candidate genes were identified through reference annotation from the cowpea reference genome *Vigna unguiculata* v1.2 available in the Phytozome database (https://phytozome.jgi.doe.gov, accessed on 28 January 2024).

## 5. Outlook

Recent studies have initiated the exploration of the complex genetic framework that governs phenotypic traits in cowpea, specifically identifying genes and genomic regions associated with variations in seed coat color and patterns. This breakthrough signifies a pivotal moment in using genetic markers to steer the breeding of cowpea varieties that meet consumer preferences and market needs. As research progresses, the focus is anticipated to shift toward harnessing these genetic insights for developing cowpea cultivars with improved seed coat attributes. This advancement is likely to incorporate marker-assisted selection (MAS) strategies to efficiently incorporate desirable seed coat traits into new varieties. The future of genetic research in cowpea coat pattern traits is promising, with ongoing studies poised to utilize genomic technologies to decode and manipulate these intricate traits, thereby propelling cowpea breeding programs forward.

## Figures and Tables

**Figure 1 plants-13-01275-f001:**
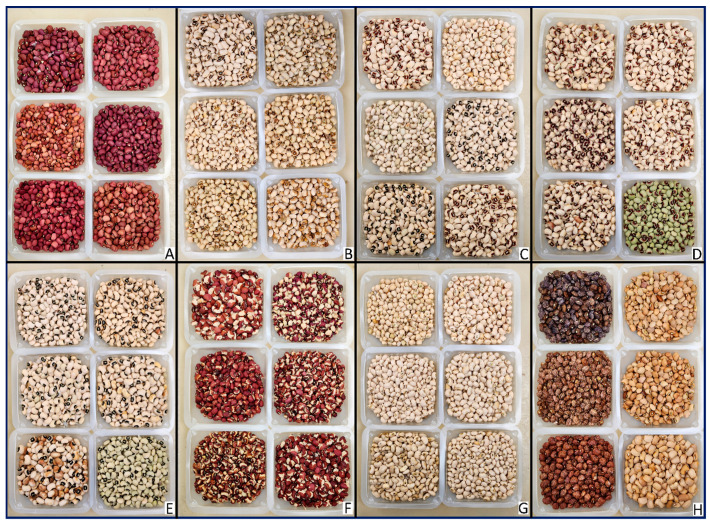
Seed coat pattern traits. Images of eight seeds patterns (**A**): Red; (**B**): Browneye; (**C**): White coat; (**D**): Pinkeye; (**E**): Blackeye; (**F**): Brown/Red Holstein; (**G**): Cream; (**H**): Brown/Tan coat.

**Figure 2 plants-13-01275-f002:**
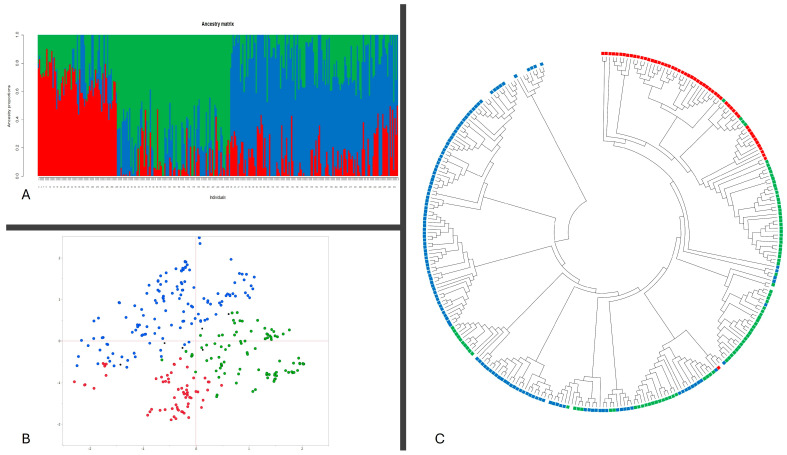
The structure, principal component, and phylogenetic analysis of 296 cowpea accessions were based on 110 K SNPs. (**A**) Classification of 296 accessions in two groups (K = 3) using LEA. The distribution of accessions to different populations is color-coded. The X-axis represents the 296 accessions, and the value on the Y-axis shows the likelihood of every individual belonging to one of the three colored subpopulations, Q1 = red, Q2 = green, Q3 = blue; (**B**) scatter diagram of PCA for 296 accessions labeled by Q groups with the colors in (**A**); (**C**) phylogenetic analysis of the accessions with the corresponding labels as Q group colors in (**A**).

**Figure 3 plants-13-01275-f003:**
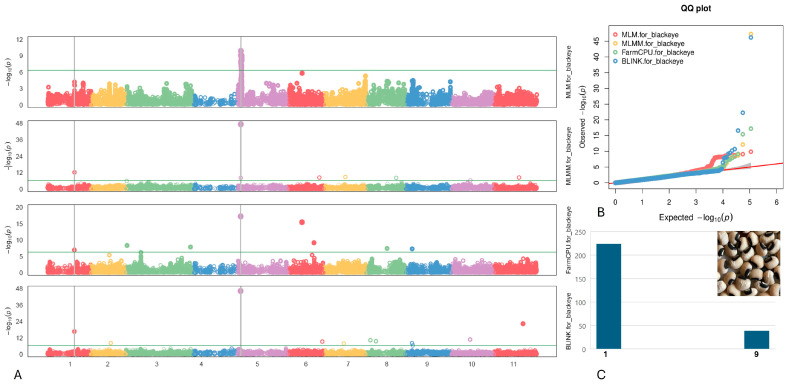
(**A**) The Manhattan plots of four GWAS models (from top to the bottom) for Blackeye pattern: Mixed Linear Model (MLM), Multiple Loci Mixed Model (MLMM), Fixed and Random Model Circulating Probability Unification (FarmCPU), and Bayesian-information and Linkage-disequilibrium Iteratively Nested Keyway (BLINK). The green transverse line of each model is the mark of significance threshold (−log(P) ≥ 6.34). The gray vertical lines mark the significant loci across multi-models. (**B**) The QQ plots of four models: MLM (red), MLMM (yellow) FarmCPU (green) and BLINK (blue). (**C**) The distribution of the Blackeye pattern in mapping panel, y-axis: counts of the accessions, x-axis: value assignment of seed coat/pattern (Blackeye = 9, non-Blackeye excluded Black Holstein and Black seed = 1).

**Figure 4 plants-13-01275-f004:**
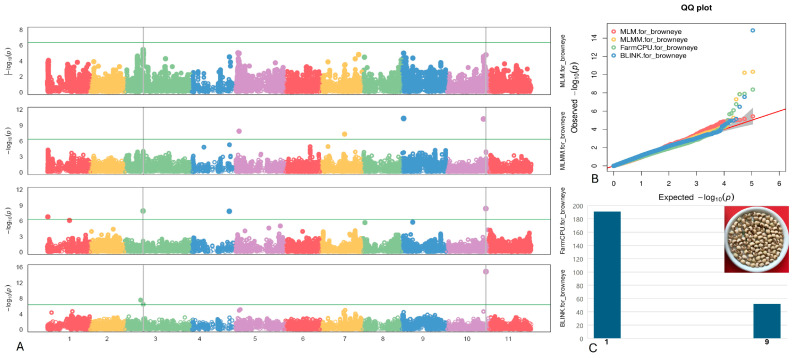
(**A**) The Manhattan plots of four GWAS models (from top to the bottom) for the Browneye pattern: MLM, MLMM, FarmCPU, and BLINK. The green transverse line of each model is the mark of significance threshold (−log(P) ≥ 6.34). The gray vertical lines mark the significant loci across multi-models. (**B**) The QQ plots of four models: MLM (red), MLMM (yellow) FarmCPU (green) and BLINK (blue). (**C**) The distribution of the Browneye pattern seeds in the mapping panel, y-axis: counts of the accessions, x-axis: value assignment of seed coat/pattern (Browneye = 9, non-Browneye excluded Brown/Red Holstein and Pinkeye = 1).

**Figure 5 plants-13-01275-f005:**
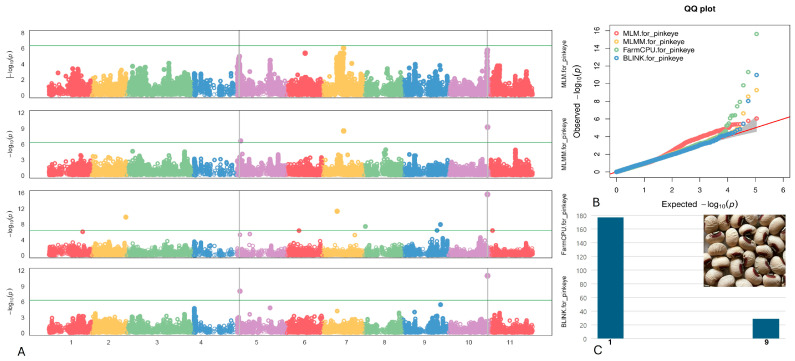
(**A**) The Manhattan plots of four GWAS models (from top to the bottom) for the Pinkeye pattern: MLM, MLMM, FarmCPU), and BLINK. The green transverse line of each model is the mark of significance threshold (−log(P) ≥ 6.34). The gray vertical lines mark the significant loci across multi-models. (**B**) The QQ plots of four models: MLM (red), MLMM (yellow) FarmCPU (green) and BLINK (blue). (**C**) The distribution of the Pinkeye pattern seeds in the mapping panel, y-axis = counts of the accessions, x-axis: value assignment of seed coat/pattern (Pinkeye = 9, non-Pinkeye excluded Brown/Red Holstein/eye and Red seed = 1).

**Figure 6 plants-13-01275-f006:**
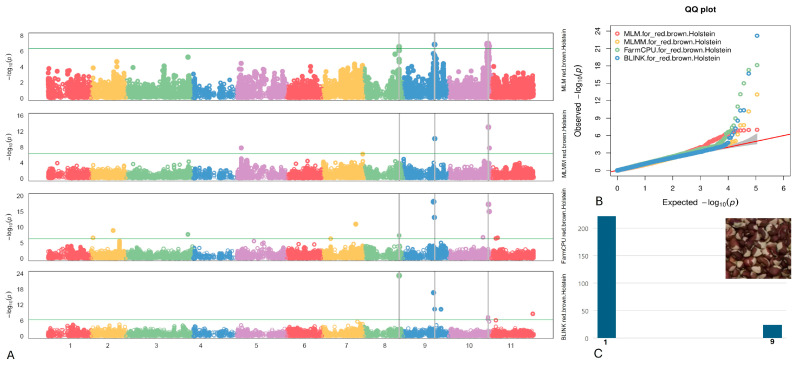
(**A**) The Manhattan plots of four GWAS models (from top to the bottom) for Red/Brow Holstein: MLM, MLMM, FarmCPU), and BLINK. The green transverse line of each model is the mark of significance threshold (−log(P) ≥ 6.34). The gray vertical lines mark the significant loci across multi-models. (**B**) The QQ plots of four models: MLM (red), MLMM (yellow) FarmCPU (green) and BLINK (blue). (**C**) The distribution of the Red/Brown Holstein pattern seeds in the mapping panel, y-axis: counts of the accessions, x-axis: value assignment of seed coat/pattern (Holstein = 9, non-Holstein excluded Pinkeye and Red seeds = 1).

**Figure 7 plants-13-01275-f007:**
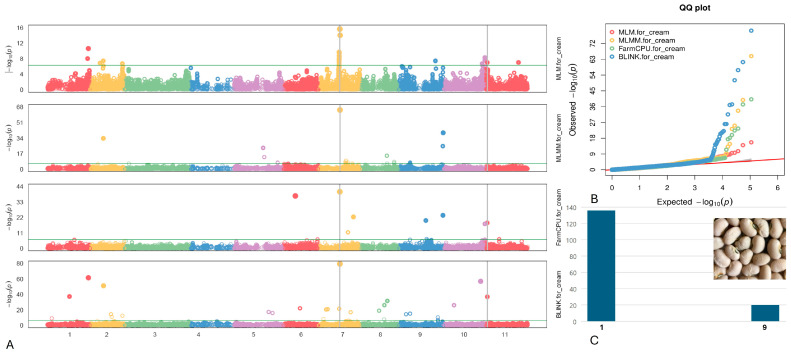
(**A**) The Manhattan plots of four GWAS models (from top to the bottom) for Cream: MLM, MLMM, FarmCPU), and BLINK. The green transverse line of each model is the mark of significance threshold (−log(P) ≥ 6.34). The gray vertical lines mark the significant loci across multi-models. (**B**) The QQ plots of four models: MLM (red), MLMM (yellow) FarmCPU (green) and BLINK (blue). (**C**) The distribution of the Cream seeds in the mapping panel, y-axis: counts of the accessions, x-axis: value assignment (Cream seed = 9, non-Cream seed excluded the White coat seeds = 1).

**Figure 8 plants-13-01275-f008:**
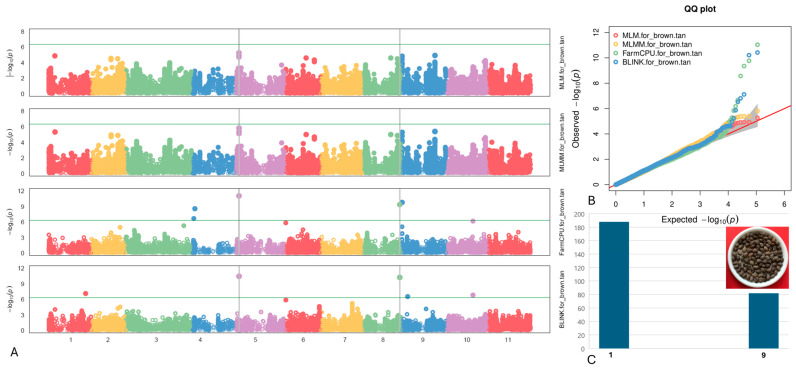
(**A**) The Manhattan plots of four GWAS models (from top to the bottom) for Brown/Tan coat: MLM, MLMM, FarmCPU), and BLINK. The green transverse line of each model is the mark of significance threshold (−log(P) ≥ 6.34). The gray vertical lines mark the significant loci across multi-models. (**B**) The QQ plots of four models: MLM (red), MLMM (yellow) FarmCPU (green) and BLINK (blue). (**C**) The distribution of the Brown/Tan seed coat in the mapping panel, y-axis: counts of the accessions, x-axis: value assignment of seed coat/pattern (Brown/Tan coat = 9, non-Brown/Tan coat excluded Red seeds = 1).

**Figure 9 plants-13-01275-f009:**
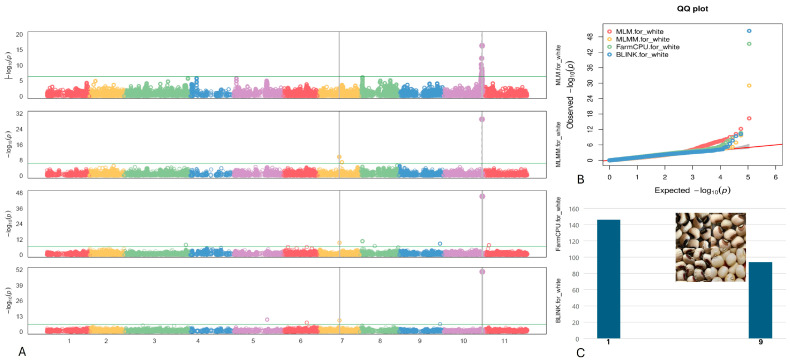
(**A**) The Manhattan plots of four GWAS models (from top to the bottom) for White coat: MLM, MLMM, FarmCPU), and BLINK. (**B**) The QQ plots of four models: MLM (red), MLMM (yellow) FarmCPU (green) and BLINK (blue). (**C**) The distribution of the White coat seeds in the mapping panel, y-axis: counts of the accessions, x-axis: value assignment of seed coat/pattern (White coat = 9, non-White coat excluded Black/Brown/Red Holstein = 1).

**Figure 10 plants-13-01275-f010:**
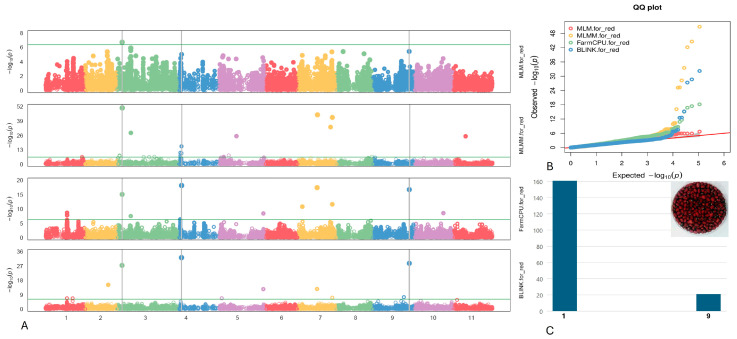
(**A**) The Manhattan plots of four GWAS models (from top to the bottom) for Red seed: MLM, MLMM, FarmCPU), and BLINK. (**B**) The QQ plots of four models: MLM (red), MLMM (yellow) FarmCPU (green) and BLINK (blue). (**C**) The distribution of the Red seeds in the mapping panel, y-axis = counts of the Red accessions, x-axis: value assignment of seed coat/pattern (Red seed = 9, non-Red seeds excluded Brown/Red Holstein and Brown/Tan coat = 1).

**Table 1 plants-13-01275-t001:** Associated SNPs and candidate genes related to the Blackeye pattern.

SNP/Gene	Chr	LOD [−log(P)] in GAPIT 3				LOD [−log(P)] in TASSEL 5		
		BLINK	FarmCPU	MLM	MLMM	SMR	GLM	MLM
Vu01_26234865	1	16.6	7.0	2.7	12.1	4.0	3.1	2.7
*Vigun01g096800*		Function: MATE efflux family protein (upstream 608 bp)						
Vu05_3015637	5	46.2	17.2	9.1	0.1	41.7	33.6	9.7
*Vigun05g037200*		Function: Cyclic nucleotide-regulated ion channel family protein (upstream 2130 bp)						
*Vigun05g037300*		Function: Calcium/lipid-binding (CaLB) phosphatase (upstream 8363 bp)						

**Table 2 plants-13-01275-t002:** Associated SNPs and candidate genes related to the Browneye pattern.

SNP/Gene	Chr	LOD [−log(P)] in GAPIT 3				LOD [−log(P)] in TASSEL 5		
		BLINK	FarmCPU	MLM	MLMM	SMR	GLM	MLM
Vu03_17993754	3	6.5	7.9	5.4	4.0	NA	7.9	NA
*Vigun03g161700*		Pentatricopeptide repeat (PPR) superfamily protein (downstream 8614 kb)						
Vu10_38465180	10	14.8	8.4	4.8	4.0	9.1	8.7	4.3
*Vigun10g165500*		Function: Transcription initiation factor TFIID subunit A (5′UTR to CDS)						

**Table 3 plants-13-01275-t003:** Associated SNPs and candidate genes related to the Pinkeye pattern.

SNP/Gene	Chr	LOD [−log(P)] in GAPIT 3				LOD [−log(P)] in TASSEL 5		
		BLINK	FarmCPU	MLM	MLMM	SMR	GLM	MLM
Vu05_3015637	5	8.0	5.3	2.5	9.3	3.0	7.4	2.1
*Vigun05g037200*		Function: Cyclic nucleotide-regulated ion channel family protein (upstream 2130 bp)						
*Vigun05g037300*		Function: Calcium/lipid-binding (CaLB) phosphatase (upstream 8363 bp)						
Vu10_38465180	10	11.0	15.6	5.8	9.3	NA	7.1	5.1
*Vigun10g165* *500*		Function: Transcription initiation factor TFIID subunit A; (5′UTR to CDS)						

**Table 4 plants-13-01275-t004:** Associated SNPs and candidate genes related to the Red/Brown Holstein pattern.

SNP/Gene	Chr	LOD [−log(P)] in GAPIT 3				LOD [−log(P)] in TASSEL 5		
		BLINK	FarmCPU	MLM	MLMM	SMR	GLM	MLM
Vu08_34097874	08	23.2	7.4	6.6	3.0	14.3	12.0	5.3
*Vigun08g170100*		Function: GDSL-like Lipase/Acylhydrolase superfamily protein (downstream 9428 kb)						
*Vigun08g170200*		Function: GDSL-like Lipase/Acylhydrolase superfamily protein (upstream 5949 kb)						
Vu09_29253230	09	16.7	18.1	5.7	4.0	15.1	15.3	5.7
*Vigun09g133400*		Function: Nucleic acid-binding, OB-fold-like protein (downstream 1343 kb)						
Vu10_39484624	10	0.3	15.1	6.7	7.8	5.4	8.7	5.5
*Vigun10g176200*		Function: Amino acid dehydrogenase family protein (downstream 289 bp)						
*Vigun10g176300*		Function: Ubiquitin-conjugating enzyme family protein (upstream 759 bp)						
*Vigun10g176400*		Function: Tudor/PWWP/MBT superfamily protein (upstream 8706 bp)						

**Table 5 plants-13-01275-t005:** Associated SNPs and candidate genes related to the Cream seed.

SNP/Gene	Chr	LOD [−log(P)] in GAPIT 3				LOD [−log(P)] in TASSEL 5		
		BLINK	FarmCPU	MLM	MLMM	SMR	GLM	MLM
Vu07_21041365	07	79.3	40.2	15.6	64.8	44.5	34.3	15.2
*Vigun07g113700*		Function: Glycosyl transferase 8 domain containing protein (upstream 526 bp)						
Vu11_1810109	11	36.9	18.2	7.0	4.0	16.2	14.4	7.1
*Vigun11g014700*		Function: Glycoside hydrolase (coding region)						

**Table 6 plants-13-01275-t006:** Associated SNPs and candidate genes related to the Brown/Tan coat.

SNP/Gene	Chr	LOD [−log(P)] in GAPIT 3				LOD [−log(P)] in TASSEL 5		
		BLINK	FarmCPU	MLM	MLMM	SMR	GLM	MLM
Vu05_3137965	5	10.42251	11.03574	5.27	5.82	6.83	5.87	4.65
*Vigun05g039300*		Function: Myb domain protein 114 (upstream 3895 bp)						
Vu08_36618860	8	10.21183	9.350665	3.57	3.78	21.33	7.41	3.91
*Vigun08g201900*		Function: Pyrimidine B (downstream 7117 kb)						
*Vigun08g202000*		Function: Co-factor for nitrate, reductase, and xanthine dehydrogenase 5 (downstream 1718 bp)						

**Table 7 plants-13-01275-t007:** Associated SNPs and candidate genes related to the White coat.

SNP/Gene	Chr	LOD [−log(P)] in GAPIT 3				LOD [−log(P)] in TASSEL 5		
		BLINK	FarmCPU	MLM	MLMM	SMR	GLM	MLM
Vu07_21041365	07	9.4	9.4	2.7	9.7	6.8	2.4	2.8
*Vigun07g113700*		Function: Glycosyl transferase 8 domain containing protein (upstream 526 bp)						
Vu10_38465180	10	50.3	45.2	16.3	29.1	48.3	23.3	19.0
*Vigun10g165* *500*		Function: Transcription initiation factor TFIID subunit A; (5′UTR to CDS)						

**Table 8 plants-13-01275-t008:** Associated SNPs and candidate genes related to the Red seed.

SNP/Gene	Chr	LOD [−log(P)] in GAPIT 3				LOD [−log(P)] in TASSEL 5		
		BLINK	FarmCPU	MLM	MLMM	SMR	GLM	MLM
Vu03_5249749	03	27.3339	15.1	6.71	50.78797	3.20	6.97	7.43
*Vigun03g063900*		Function: Phosphatidylinositol 4-OH kinase beta1 (downstream 3160 bp)						
*Vigun03g064000*		Pyridoxal phosphate-dependent transferase (upstream 1694 bp)						
*Vigun03g064100*		Pyridoxal phosphate-dependent transferase (upstream 6901 bp)						
Vu04_2848654	04	32.18	18.15	5.01	8.34	20.59	11.47	6.34
* Vigun04g034400 *		Pentatricopeptide repeat (PPR-like) superfamily protein (downstream 5705 bp)						
* Vigun04g034500 *		Knotted-like homeobox (KNOX) (upstream 7031 bp)						
Vu09_38749459	09	28.64381	16.7371471	5.43	2.1	10.25	10.60	5.41
*Vigun09g213300*		Function: Alginate regulatory protein AlgP (downstream 5897 bp)						
*Vigun09g213400*		Function: SERINE-THREONINE PROTEIN KINASE (upstream 6104 kb)						

## Data Availability

The data that support the findings of this study are available in the Appendix A. The SNP data are available in FigShare: 10.6084/m9.figshare.25650930.

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
