# Peer review of "Genetic Dissection of Diverse Seed Coat Patterns in Cowpea through a Comprehensive GWAS Approach"

_plants, 2024, doi:10.3390/plants13091275_

Round 1
Reviewer 1 Report
Comments and Suggestions for Authors
This manuscript shows general fluency of English writing with minor grammar check and rephrasing required.
Author Response
1 This study depicted a scenario of using GWAS in identifying potential associated loci on seed coat patterns in cowpea collection. The author applied multiple association models in digging the genetic components correlated to demanding phenotypic traits. Analyses were aiming to increase the understanding of the critical value of the cowpea seed collection in studying the genetic variation of a candidate crop species in agriculture. Authors also raised up the advantage of using a diverse group and genome wide approaches in assisting the functional genomic study on valuable economic traits. I have several major concerns about this manuscript. Personally, it is worthwhile to address/explain these points to increase the research value of this study as they may be more convinced to general readers in this field.
- This manuscript placed GWAS as the core approach to understand the SNP association to specific functional genes in cowpea genome. However, I found the detailed method of both SNP calling and GWAS workflow is vaguely described in the methods. This type of in silico study normally requires a certain level of repeatability since it claimed be a citable approach for future studies. The raw sequencing data as well as the analytical pipelines should be clear and easy to repeat. I would like to request the detailed information about but not limited to how the resequencing reads were mapped onto the reference genome, the SNP calling processes, and the GWAS parameter choice. Please revise the methodology section of this manuscript.
In detail:
Reviewer’s comment: Line 98-99: Is the SNP number exactly 110k (110,000)? If not, to be precise, either having the exact number of SNPs or using approximating phrases instead.
Line 103-104: Please add the reference genome size here. It would be comparable across organisms when mentioning the inter-SNP distance.
Line 441-442: How many grams of frozen leaf tissue were used in DNA extraction?
Line 450-452: What are the methods for SNP screening? Missing those steps would not form a convincing SNP calling results. Please add those critical information with detailed software, parameter choices, and the filtering criteria.
Reply: Thank you for your professional suggestions. We re-wrote the Mothed section of “4.2. DNA extraction and whole genome sequencing”, all the necessary information is included.
Reviewer’s comment: Line 466: Please specify how did you operating the software STRUCTURE.
Reply: Thank you for the suggestion, actually we are doing the structure analysis by LEA package in R. We are sorry for this confusion, and we added more information in this part.
Reviewer’s comment: Line 467: What is the software for the phylogenetic trees and which and why the model was chosen.
Reply: Thank you for reminding us. We re-wrote this paragraph to eliminate confusion and more necessary information was added.
Reviewer’s comment: Line 470-476: Please offer the detailed pipeline/code for GWAS study. The current description is more like a list of software and packages.
Reply: Thank you for your kind reminding, we have added this information in the MS.
Reviewer’s comment: Line 524-525: The Supplementary Material for reviewing does not include sufficient data. The expected data for this study should include the reads and variant calling files. I strongly encourage those data being uploaded to public databases before the final revision.
Reply: Thank you for reminding! We have uploaded the genotype data to the: 10.6084/m9.figshare.25650930
- Interpretation of the population structure analysis need to be revised.
Line 106-120, and Figure 2 described the population structure of the 296 cowpea accessions. The results seemingly suggested three subgroups from a previous mixed grouping. However, analyses had unattended odds to clearly draw the conclusion.
2 In detail:
Reviewer’s comment: Line 106-109: What is the method and the result for determining the optimal K is 3 in your STRUCTURE analysis? Does the input data satisfy with the requirement of the Admixture model?
Reply:
1) Yes, the method for calculate the best K was described in method section.
2) Yes, our input data is fully satisfying with the LEA package we used for the structure analysis and the Admixture was generated by the structure result.
Reviewer’s comment: Line 113-114: How to explain the incongruence between the geological categorization and the grouping using the STRUCTURE?
Reply: Geological categorization is based on the physical locations where the accessions are collected, which may not always reflect the genetic makeup of those populations due to historical migration, gene flow, and the presence of cryptic genetic structures. On the other hand, STRUCTURE analysis relies on genetic data to identify populations or groups within a dataset based on allele frequencies, which can reveal hidden genetic differentiation not observable through geographical categorization alone. The observed incongruence could be due to several factors as: gene flow between populations from different geological regions; the diverse genetic structures in the single geological population; environmental pressures and adaptive evolution.
However, it's important to clarify that the scope of our manuscript does not extend to geological categorizations. Our study focuses solely on the genetic basis of seed coat color and pattern diversity in cowpea, utilizing a GWA approach. Therefore, geological factors weren't part of our analytical framework in this manuscript. We're grateful for your observation and appreciate your understanding.
Reviewer’s comment: Line 114-115 and Fig2b: PCA results seems to be unclear. No grouping method supports the conclusion that there are three distinct groups. Additionally, PCA plot missing the axis information and the variation represented by each plotting principal component.
Reply: We appreciate your insightful comments regarding the PCA results. In acknowledgment of your observations, it's important to clarify that the PCA on its own does not offer irrefutable evidence for the existence of three distinct groups. The classification into three groups is primarily informed by our structure analysis, highlighted by three different colors representing the structure results. Our objective was to validate these classifications by integrating the structure analysis outcomes with the PCA plot, facilitating a visual confirmation of distinct group separations. This method allowed us to clearly delineate the groups within the PCA plot, thereby demonstrating the consistency between the two distinct analytical approaches. We apologize for any ambiguity caused by the omission of axis details and the percentage of variation each principal component accounts for in the PCA plot. We have re-did the figure to include this crucial information for this analysis.
Reviewer’s comment: Line 118-120: The Figure S1 is not the kinship matrix. Please add the correct figure since it is important to know the kinship result in this low heterozygosity data. (Seemingly super inbred within groups).
Reply: We sincerely apologize for the oversight regarding Figure S1 and appreciate for pointing out this issue, and we have fixed this it.
Reviewer’s comment: Fig2c: If the phylogeny is rooted properly, it looks like that none of the three groups is monophyletic nor exclusively forming clades. No distinct common ancestors were found for each color group, i.e., from the tree it is hard to follow the idea that there are three groups in your entry dataset. Please consider either interpret the phylogeny along with the other analyses or apply different tree reconstruction methods and evolutionary modeling rather than a NJ tree.
Reply: We appreciate your insights regarding the phylogenetic analysis presented in our manuscript. Indeed, the Neighbor-Joining (NJ) tree method has its limitations in ancestry analysis, yet its utility in phylogenetics remains significant. Our primary intention in incorporating phylogenetic analysis was to complement our structure analysis, aiming to demonstrate consistency across different analytical approaches. Based on the outcomes, we believe the results align well with our objectives. Given that the focus of our paper is on GWAS rather than comprehensive ancestry analysis, we opted for a balanced approach that supports our GWAS findings without delving deeply into ancestry complexities. This choice was made to maintain the manuscript's clarity and focus, avoiding an overly extensive exploration of ancestry that might dilute the core message of our GWAS findings. We hope this explanation clarifies our methodological decisions and their alignment with the study's goals.
- All results of the GWAS are generally straightforward to me. Nevertheless, adding details and future explanations can be more informative to followed up studies. Since the GWAS results sections are formatting similarly, I will select one of the traits as an example to address some concerns unless specified. Please check all the GWAS results that share the similar structure in the text and figures.
In details:
Reviewer’s comment: Fig3-10: Please add sufficient descriptions to the figure legend. For Fig3-10a, the outside ring I interpreted as the reference genome loci. Please describe the green/white/red lines on each chromosome. The cowpea has 2n=22 chromosomes while the ring has an additional “chromosome” without caption. I suspect it is the unattached scaffolds/contigs in genome assembly. Why there is no SNP call on these pieces? There is a potential of missing data from those sequences if not included.
Reply: Thank you for your thorough checking and constructive feedback of the Figures. You correctly identified issues in Fig A, prompting us to meticulously double checked our raw data and calculations, particularly concerning the “unattached scaffolds/contigs” as you suggested.
Upon our check, we discovered that the issues did not from missing data or unaccounted genomic elements, but rather from a visualization error produced by the software circular chart generation; this anomaly was not present in other types of Manhattan plots. Regrettably, the precise cause of this visualization error remains unidentified at this juncture. We replaced the problematic circular images in all the figs and hope this adjustment satisfactorily resolves the issue and enhances the clarity and integrity of our work.
Reviewer’s comment: Fig3-10a: The inner rings have the dashed red lines as the thresholds of claiming significance. Are those adjusted Bonferroni correction at certain alpha value? Please add those in figure legends.
Reply: Yes, you're correct. The transverse lines in Fig3-10a represent the thresholds for claiming significance, and they are indeed adjusted using Bonferroni correction at a certain alpha value, and we added this information in the Method section and Fig descriptions. Specifically, all the models depicted in the figure utilize the same Bonferroni correction threshold.
Reviewer’s comment: Fig3-10c: The description of these figure is hard to follow. Can you explain it better? What is the information you would like to offer to readers?
Reply: Thank you for reminding us, we have modified the descriptions.
Reviewer’s comment: Line160-162: What is the structural location of these SNPs? Are these located in the exons/introns/intergenic regions? I think these are critical to future work when specific traits being studied. Table1-8: For the -logP values in the table, which of them are statistically significant in each model? Please add this information since the threshold seems to be different in your models.
Reply: Thank you for your inquiry regarding the significance of the -logP values in Table1-8. It's important to clarify that we applied the same significance threshold (-log(P) ≥ 6.34) across all models in our GWAS analysis. However, we also considered the consistency of SNP significance among the various models we employed. Specifically, out of the seven models utilized, only those SNPs that showed significance in more than four models were considered as significant. This approach ensures robustness and reliability in identifying SNPs associated with the trait under investigation. We rewrote this approach in line We added more information about the SNP location with the genes, in table Table1-8. And further discussions on this topic have been included in the Discussion section of our study. We believe these updates enhance the comprehensibility and reliability of our findings for future research endeavors as you are concerned.
Reviewer’s comment: Table1-8: If the SNP is closed to more than one gene within the 5kb window, does it indicate the SNP is located in an intergenic region? Table1-8: If there is no functional annotation of a specific SNP, does it mean there is no annotation for specific region, or it is located in an intergenic region? If it is an unknown gene, does it have homologs in Arabidopsis thaliana?
Reply: Thank you for your professional suggestions, we added more information and re-wrote the results and discussion sections.
Please address these major concerns above while revising the manuscript.
Minors: Please fix all the software citations and version information not limited to the listed below. Please make sure all the species names are italicized.
Reviewer’s comment: Line 64-70: Rephrase the sentences? Line72-74: Two studies but only one citation?
Reply: Thank you for reminding! We modified this from line 67-76.
Reviewer’s comment: Line106-109: These information of the package and software should be moved to methods. There are missing references for LEA packages and R (R core and/or R studio).
Reply: Thank you! We added the reference!
Reviewer’s comment: Line 119-120: The phrase “additional insight” here is vague. What is the purpose of the kinship matrix in diversity study?
Reply: Thank you! We corrected this obscure and described the usage of kinship in line 122-124.
Reviewer’s comment: Line130-133: What is the purpose of estimating LD and LD decay in your study? Is there any comparable study to show the interesting pattern of your dataset?
Reply: Thank you for your inquiry about the purpose of estimating LD decay in our study. Estimating LD is crucial as it helps us understand the non-random association of alleles at different loci, which is pivotal for identifying genetic markers linked to traits of interest in genome-wide association studies. In early GWAS studies, we determined the sufficiency of marker density through LD decay. Nowadays, due to the increase in the number of markers and a common understanding of the genetic decay distance for specific species, many studies do not necessarily perform LD analysis. This study specifically describes this to ensure the integrality of the data, and the results are as expected.
Reviewer’s comment: Line138: What is the alpha value of the Bonferroni correction in your study?
Reply: We added this information in Mothed section.
Reviewer’s comment: Line 319-324: Need the source of these statements. Also, solely based on this information, I found there is limited evidence for genetic variance in your data. A phenotypic variance suggesting the genetic variance might be clearer. A genetic diversity usually requires rigorous population genetics test on several parameters.
Reply: Thank you for your comment. I apologize for any confusion in the description of our methods. We have conducted the relevant analyses in several previous studies, which we have now referenced in our paper to substantiate these statements.
Reviewer’s comment: Line347: What does “a decay distance of 87.6Kb” mean?
Reply: The distance within which the correlation between genetic markers decreases to a certain level, in this case, spanning 87.6 Kb on a chromosome. It means that in the population studied, alleles at loci that are less than 87.6Kb apart are likely to be inherited together due to their proximity on the chromosome.
Reviewer’s comment: Line359-360: Personally, I would be vigilant to the word like “nuance” in the text when claiming an established approach in the manuscript. Both GWAS and genetic mapping methods were widely applied in a lot of studies.
Reply: Thank you for your consideration. We modified these words.
Reviewer’s comment: Line378-380: Citations?
Reply: We added the reference.
Reviewer’s comment: Line382: Which S Table is referred here? Line450-452: Any references for these criteria?
Reply: We added the references.
Reviewer’s comment: Line454: No citation for R. Line464: No citations for software.Line466: No citations for STRUCTURE. Line467: No software selection for phylogeny. Line468: No citations for TASSEL.
Reply: Thank you! This paragraph was modified and the citations were added.
Reviewer’s comment: Figure S1: There are white blocks seems not aligned with the main bar (e.g. on Chr04). Could you either explain or fix them? Also, the color scale selection is started at 1 with a step length of 1925, which is a very random number. Does this represent a range of SNP density in a range? Why not using uniformed ranges like 1-2000, 2001-4000 etc. instead?
Reply: Thank you. We replaced this figure.
Reviewer’s comment: Figure S2: The LD decay plot is an averaged R-square decreasing while the pairwise physical distance increases. Please specify your data dots on the plot are pairwise R-square or the mean Rsquare. Also, please fix the R2 in the title and the main text using correct superscripts.
Thank you! We modified the figure description as required.
Reviewer’s comment: Figure S3-5: Remove the software information in the title of figures. Also, even the figures seems to be direct outputs from the software, it still need the axis information and better with the thresholds onto. The resolution of these figures should improve as well.
Reply: Thank you! We modified and re-did the figures as required.

Reviewer 2 Report
Comments and Suggestions for Authors
Seed coat pattern in cowpeas is an interesting trait in consumer market and breeding practice. This paper presents a genetic mapping study on seed coat pattern/color variation in a diverse cowpea panel with the aim to find SNPs and putative candidate genes responsible for specific seed patterns.
In short, I hope the authors could have an informative study with vigorous methodologies and their insights to enrich the readership and really advance future research in this field, instead of simply listing these dry results. That being said, my main concern is the repetitive presentation of mapping results of these patterns without providing clear genotyping and phenotyping information. The boxplots (Fig. 3-9C) of color/pattern distribution are poorly presented. What does “ x-axis = value assignment of target/non- target traits” mean? The simple presence/absence of a specific pattern among panel accessions is not informative for quantitative association study.
Other comments,
Abstract reads more like a summary of the results.
Line 48-50 Please provide references for these claims on color preference.
Line 51-52 What do “enhancing agricultural methodologies” refer to?
Please refrain from the use of subjective words “captivating”, “intriguing”, “fascinating” etc. in scientific writing.
Comments on the Quality of English LanguageExtensive revision is needed.
Author Response
Reviewer’s comment :Seed coat pattern in cowpeas is an interesting trait in consumer market and breeding practice. This paper presents a genetic mapping study on seed coat pattern/color variation in a diverse cowpea panel with the aim to find SNPs and putative candidate genes responsible for specific seed patterns. In short, I hope the authors could have an informative study with vigorous methodologies and their insights to enrich the readership and really advance future research in this field, instead of simply listing these dry results. That being said, my main concern is the repetitive presentation of mapping results of these patterns without providing clear genotyping and phenotyping information. The boxplots (Fig. 3-9C) of color/pattern distribution are poorly presented. What does “ x-axis = value assignment of target/non- target traits” mean? The simple presence/absence of a specific pattern among panel accessions is not informative for quantitative association study.
Reply: Thank you for your thoughtful feedback. We appreciate your interest in the robustness of the methodologies used in our study on seed coat pattern variation in cowpea and understand your concerns about the clarity and depth of the genetic mapping results presented. In our study, we assigned values of 9 for target traits and 1 for non-target traits as a standard practice to categorize typical quality characteristics. Some studies use “0” and “1” assignments, while it would not impact the result of the GWAS. We apologize for not clarifying the non-target patterns previously. Due to the potential for confusion between similar seed patterns, such as "red seed" and "brown/tan coat," we carefully excluded seed coat types that could be mistaken for one another during our analyses. For instance, when analyzing "cream seed" traits, all white coat types were excluded to avoid confusion. Conversely, "white coat" analyses included "cream seed" types, considering them as white coats without distinct patterns. This approach helps to refine our quantitative association study by minimizing misclassification. We added this information in the Fig3-10.
Other comments,
Abstract reads more like a summary of the results.
Line 48-50 Please provide references for these claims on color preference.
Line 51-52 What do “enhancing agricultural methodologies” refer to?
Please refrain from the use of subjective words “captivating”, “intriguing”, “fascinating” etc. in scientific writing.
Reply: Thank you for your valuable suggestions. Per your request, we have thoroughly revised several sections of the manuscript, including, but not limited to, the highlighted parts. We are hopeful that these revisions make the paper more informative and impactful for the readership. Thank you once again for helping us improve our work.

Round 2
Reviewer 1 Report
Comments and Suggestions for Authors
Thanks to the author for the careful revision of this manuscript. I am satisfied with all the corrections with no further comments. The decision can be made by the editors.
Author Response
We deeply appreciate your comments and suggestions, which have significantly enhanced our manuscript.
Reviewer 2 Report
Comments and Suggestions for Authors
Thanks for the authors' response. Unfortunately, I found they are either too general or not addressing the review comments. For instance, I didn't find any specific point-by-point response or highlighted text which directly address my previous comment on those poorly presented boxplots (Fig. 3-9C) and the followings,
Abstract reads more like a summary of the results.
Line 48-50 Please provide references for these claims on color preference.
Line 51-52 What do “enhancing agricultural methodologies” refer to?
Comments on the Quality of English Language
See above,
Author Response
Reviewer’s comment: Thanks for the authors' response. Unfortunately, I found they are either too general or not addressing the review comments. For instance, I didn't find any specific point-by-point response or highlighted text which directly address my previous comment on those poorly presented boxplots (Fig. 3-9C) and the followings,
Reply: Sorry for the misunderstanding of the last revision, we have re-made all the Figs especially for the boxplots this time.
Reviewer’s comment: Abstract reads more like a summary of the results.
Reply: Thank you, we re-wrote the Abstract.
Reviewer’s comment: Line 48-50 Please provide references for these claims on color preference.
Reply: Thank you for reminding! We added the references.
Reviewer’s comment: Line 51-52 What do “enhancing agricultural methodologies” refer to?
Reply: Thank you for the suggestion. We found this sentence is ambiguous. We re-wrote this sentence.
Round 3
Reviewer 2 Report
Comments and Suggestions for Authors
Thanks for the response. I don't have further comments.
Comments on the Quality of English LanguageI suggest further editing to improve the structure and syntax of some sentences.